# SPC: Evolving Self-Play Critic via Adversarial Games for LLM Reasoning

**Jiaqi Chen**[1]  **Bang Zhang**[*]  **Ruotian Ma**[2†]  **Peisong Wang**[3]
**Xiaodan Liang**[4]  **Zhaopeng Tu**[2]  **Xiaolong Li**[2]  **Kwan-Yee K. Wong**[1†]
[1]The University of Hong Kong  [2]Tencent
[3]Tsinghua University  [4]MBZUAI
Project: https://chen-judge.github.io/SPC/

## Abstract

Evaluating the step-by-step reliability of large language model (LLM) reasoning, such as Chain-of-Thought, remains challenging due to the difficulty and cost of obtaining high-quality step-level supervision. In this paper, we introduce **S**elf-**P**lay **C**ritic (**SPC**), a novel approach where a critic model evolves its ability to assess reasoning steps through adversarial self-play games, eliminating the need for manual step-level annotation. SPC involves fine-tuning two copies of a base model to play two roles, namely a "sneaky generator" that deliberately produces erroneous steps designed to be difficult to detect, and a "critic" that analyzes the correctness of reasoning steps. These two models engage in an adversarial game in which the generator aims to fool the critic, while the critic model seeks to identify the generator's errors. Using reinforcement learning based on the game outcomes, the models iteratively improve; the winner of each confrontation receives a positive reward and the loser receives a negative reward, driving continuous self-evolution. Experiments on three reasoning process benchmarks (ProcessBench, PRM800K, DeltaBench) demonstrate that our SPC progressively enhances its error detection capabilities (e.g., accuracy increases from 70.8% to 77.7% on ProcessBench) and surpasses strong baselines, including distilled R1 model. Furthermore, SPC can guide the test-time search of diverse LLMs and significantly improve their mathematical reasoning performance on MATH500 and AIME2024, surpassing those guided by state-of-the-art process reward models.

## 1 Introduction

The Chain-of-Thought (CoT) [1–3] reasoning process, which emerges in the autoregressive generation of large language models (LLMs), has been applied to address a variety of complex tasks [4–13]. Training methods such as Supervised Fine-Tuning (SFT) [14, 15], Reinforcement Learning from Human Feedback (RLHF) [16, 17], and self-play reinforcement learning [18, 19], have demonstrated success in obtaining high-quality CoT. Recently, the popular o1 [20], R1 [21], and QwQ [22] utilize large-scale reinforcement learning for training and employ test-time scaling to generate long CoT, further enhancing their reasoning capabilities. As the CoT generated by LLMs becomes increasingly complex and diverse, it is particularly important to verify the reliability of the reasoning process, analyze the potential errors in reasoning steps, and guide the test-time search to improve the reasoning process [23–31].

A number of verification models have been developed to analyze and evaluate the reasoning process of LLMs. For example, outcome verifiers [25] provide outcome-level validation to rank or reward multiple responses from LLMs. Process verifiers [23, 25], which validate each step in the reasoning

---

[*]Independent researcher. [†]Corresponding authors.

39th Conference on Neural Information Processing Systems (NeurIPS 2025).

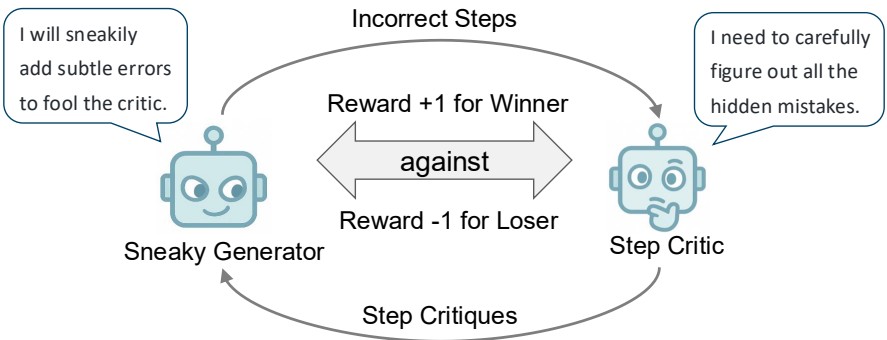

Figure 1: We continuously generate reinforcement training samples for the critic through adversarial games. The sneaky generator aims to create subtle erroneous steps to challenge the critic, while the critic must accurately distinguish between correct and incorrect steps from a mixed input of them. Benefiting from the opposing optimization objectives, both models can evolutionally learn from each other, akin to how humans improve their skills in board games through competition.

process, have proven crucial in recent advances in LLM reasoning [26, 32–34]. However, there are several challenges that limit the development of such step-level approaches. Firstly, while it is relatively simple to extract the final predicted answer and determine the correctness of a solution, determining the correctness of a reasoning step and automatically obtaining well-annotated step data for training a process verifier is much more difficult. Secondly, LLMs are updated rapidly, and heavy human expert annotations on the outputs of specific LLMs may not be applicable to the latest LLMs due to distributional differences. Thirdly, the dataset limited to step correctness annotations restricts the training of a critic model – preventing it from providing substantive feedback and reducing it to merely a scoring mechanism for verification.

In this paper, we introduce a novel **S**elf-**P**lay **C**ritic (**SPC**) to diagnose potential errors and provide valuable critiques for each step in the mathematical reasoning process. Inspired by the self-play framework [19], we propose an adversarial game between a sneaky generator and a critic to continuously generate samples for reinforcement learning, thereby evolving the capabilities of the critic model. Specifically, we first employ supervised fine-tuning to initialize a base model as a sneaky generator, converting correct steps into incorrect steps that can significantly impact the success rate of problem-solving. Concurrently, we initialize an identical base model to play the role of a critic, whose goal is to identify the correctness of these reasoning steps and provide some critiques for them. As shown in Fig. 1, we put these two models in an adversarial game by feeding the incorrect steps successfully generated by the sneaky generator to the critic. Through this adversarial game, we anticipate that the sneaky generator can simulate errors that can practically influence the reasoning of LLMs while remaining difficult for the critic to detect. On the other hand, the critic is expected to gradually address its shortcomings and improve its ability to catch all errors in the reasoning steps. Benefiting from this design, we continuously generate positive/negative samples from different LLMs for reinforcement learning without the need for additional human annotations, facilitating the iterative evolution of a critic model which can provide valuable step critiques.

Extensive experiments have been conducted to validate the effectiveness of our proposed self-play critic. After one round of supervised fine-tuning on Qwen2.5-7B-Instruct and two rounds of iterative reinforcement fine-tuning, our SPC has shown continuously evolving performance on three human-annotated reasoning process assessment benchmarks (ProcessBench [27], PRM800K [23] and DeltaBench [35]). For instance, the average accuracy of SPC on PRM800K has gradually improved from 71.0% to 75.8%, surpassing the 71.4% performance of the same-sized distilled model of R1 [21]. We further introduce a new approach to utilize our tailored critic model, wherein the critic predicts the correctness of each step during LLMs' test-time search. This allows the LLM to promptly abandon incorrect steps and regenerate new steps, rather than waiting until the entire solutions are generated and then scoring them using verifiers. Experiments on MATH500 [36] and AIME2024 [37] indicate that SPC can enhance mathematical reasoning for three different types of LLMs, including popular Llama [38], Qwen [12], and distilled R1 [21] with long CoT reasoning process.

## 2 Related Work

**LLM Reasoning** Powerful large language models (LLMs) [4, 5, 7–13] are becoming increasingly adept at constructing Chain-of-Thought (CoT) to tackle complex reasoning tasks, such as solving math problems and code generation. The recently popular o1 [20], R1 [21], and QwQ [22] models are further equipped with exceptional deep thinking capabilities, allowing them to construct long CoT during inference to decompose complex tasks and even perform extensive self-critique and self-correction [39–41]. However, fine-grained analyses in a recent research [35] indicate that the effective proportion of self-critique in these long CoT is still very low, and biases exist in the self-critique of their own reasoning processes. It is therefore necessary to have a simple external critic for assessing the reasoning steps of various LLMs, providing step-level critiques.

**Verification and Critique for LLM** Verifiers [23–27] can enhance reasoning performance by ranking or integrating multiple responses generated by LLMs during inference. Additionally, they can also provide more accurate rewards during training to guide the optimization of LLMs. Verifiers can be categorized into two types, namely outcome reward models (ORMs) and process reward models (PRMs). ORMs provide solution-level scores for the entire problem-solving process, whereas PRMs assign step-level scores to each reasoning step, which can be aggregated to produce a more accurate solution-level score. Recent works [28–30, 42] propose critic models for verification, arguing that scalar scores have limited ability in evaluating the outputs of LLMs. In contrast, feedback in natural language form can activate the thinking capabilities of LLMs, resulting in more reliable critiques to represent the correctness of reasoning. In this work, we explore how to analyze the correctness of the current step and provide step-level critiques based on partial reasoning steps.

**Self-Play** Self-play [43, 44] is a method in reinforcement learning where an agent interacts with several copies of itself in an environment to learn specific actions. A significant advancement in self-play is demonstrated by AlphaGo [45] and AlphaZero [46], which greatly surpass human champions in the game of Go, without the need for human knowledge in training. Recent studies apply self-play to LLM alignment and enhancement [18, 19, 47–50]. For example, Kirchner et al. [19] proposed a solution-level game between a powerful prover and a weak scoring verifier to enhance the legibility of the LLM, though resulting in performance degradation. Cheng et al. [47] introduced a Taboo language game between an attacker and a defender to improve LLMs' reasoning abilities. In this paper, we design an adversarial game to generate data for training a step-level critic, which provides correctness analysis for the reasoning steps of LLMs.

## 3 Methodology

### 3.1 Overview

Training a step-level critic requires a large amount of data annotated with step correctness. However, collecting step-level data presents considerable challenges. First, identifying and annotating the reasoning errors of powerful LLMs requires professionals with relevant expertise. Second, LLMs are rapidly updated, and the labor-intensive annotations may become outdated and inapplicable to the latest LLMs due to distributional shifts. Third, there is no definite answer for each step, complicating the definition of "incorrect" and the automation of the annotation process.

In this work, we design a self-play framework to enable the self-evolution of step-level critic by automatically producing step-level annotation through an adversarial game. As shown in Fig. 2, our framework involves two opposing models, i.e., a sneaky generator $S$ and a step-level critic $C$.

**Sneaky generator** converts correct reasoning steps from LLMs into incorrect ones, automating the creation of numerous steps with potential errors. Its goal is to generate sneaky steps that not only decrease the reasoning success rate of LLMs but also deceive the critic (i.e., the critic fails to detect the erroneous steps). Specifically, given a problem $p$ and a correct partial reasoning trajectory $\tau_{:k} = (t_1, t_2, ...t_k)$ produced by an LLM solver, the sneaky generator $S$ converts the last correct step $t_k^c$ into a *candidate sneaky step* $t_k^i = S(p, \tau_{:k-1}, t_k^c)$. This candidate becomes a *valid sneaky step* $t_k^i$ if it significantly impacts the solver's success rate in the subsequent completion from this step. If the sneaky step is invalid or the critic detects the errors in a valid sneaky step, the sneaky generator receives a negative reward. Conversely, if the critic fails to detect the errors in a valid

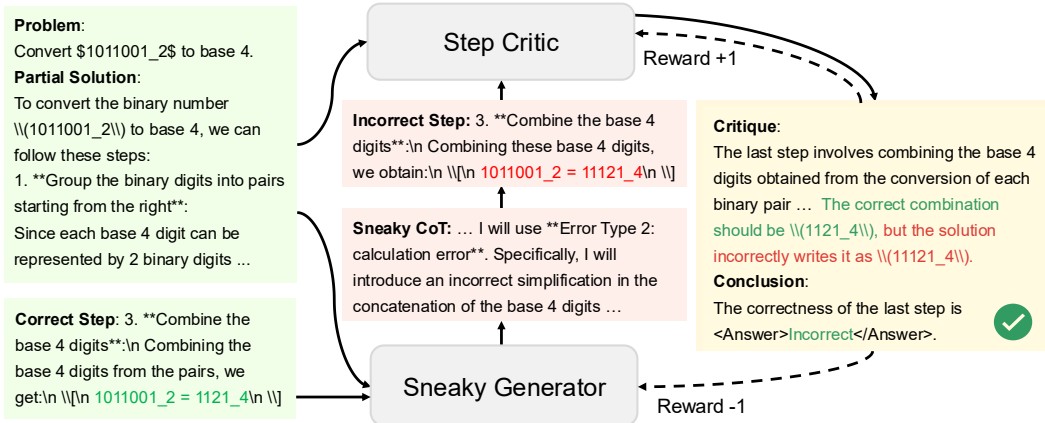

Figure 2: The framework of our proposed SPC. We randomly select a correct step along with the partial solution before that step and feed them into the sneaky generator, which first selects one of the predefined error types and then converts the correct step into an incorrect step. The successfully generated incorrect step is then fed to the critic for error detection. If the critic successfully identifies the error, it receives a reward of +1, while the sneaky generator incurs a reward of -1. If the critic is deceived, the critic and sneaky generator are rewarded -1 and +1, respectively.

sneaky step, the sneaky generator receives a positive reward. By automatically evaluating the success of the generated sneaky steps, we then employ reinforcement learning to enable the self-evolution of the sneaky generator.

**Step Critic** aims to identify all potential errors in the reasoning steps of LLMs. In each iteration of the adversarial game, the critic's role is to detect all error steps generated by the sneaky generator. Specifically, given a partial reasoning trajectory $\tau_{:k-1} = (t_1, t_2, ...t_{k-1})$ and a valid sneaky step $\overline{t_k^i}$ produced by the sneaky generator $S$, the critic $C$ is expected to identify $\overline{t_k^i}$ by generating a step-level critique. The success or failure of detecting the sneaky step determines the critic's rewards, allowing continuous optimization through reinforcement learning.

Overall, these two models have opposing objectives, allowing them to evolve through adversarial self-play. In the following sections, we explain how to initialize these models and continuously generate positive and negative samples for reinforcement learning through adversarial games.

## 3.2 Initializing Sneaky Generator

To initialize the sneaky generator $S_0$, we train the base model Qwen2.5-7B-Instruct [12] using Supervised Fine-Tuning (SFT) to equip it with the fundamental capability to generate incorrect steps. To ensure the accuracy of the initialization data, we use correct-incorrect step pairs from PRM800K [23] to construct an error step transformation process. Specifically, we extract correct-incorrect step pairs $< t_k^c, t_k^i >$ with the same problem $p$ and partial solution $\tau_{:k-1} = (t_1, t_2, ...t_{k-1})$ (the steps preceding the extracted pairs) from PRM800K. We next prompt GPT-4 to create a chain-of-thought transformation $\mathcal{T}_{\text{CoT}}(t_k^c) \rightarrow t_k^i$ by first selecting an error type from five predefined common error types (see Sec. B) and then performing a detailed transformation. This process results in a transformation behavior cloning dataset $(\mathbf{x}, \mathbf{y}) \sim \mathcal{D}_{bc}^S$, where $\mathbf{x} = (p, \tau_{:k-1}, t_k^c)$ as input, $\mathbf{y} = \mathcal{T}_{\text{CoT}}(t_k^c) \rightarrow t_k^i$ as output. We then finetune Qwen2.5-7B-Instruct on dataset $\mathcal{D}_{bc}^S$ to obtain a policy $\pi_\theta$ for the initial sneaky generator $S_0$ using the SFT loss:

$$\mathcal{L}_{\text{SFT}} = -\mathbb{E}_{(\mathbf{x}, \mathbf{y}) \sim \mathcal{D}_{bc}^S}[\log \pi_\theta(\mathbf{y}|\mathbf{x})]. \tag{1}$$

**Automated Validation for Sneaky Generator**  To form an adversarial game, we need to annotate the generated steps and feed actual incorrect steps to the critic model. However, existing LLM-as-a-Judge methods [35, 51] inevitably introduce bias, while the manual annotation is excessively labor-intensive. We therefore propose evaluating the impact of different steps on the problem-solving success rate to ascertain whether a sneaky step can be considered incorrect. Concretely, based on a

correct solution generated by an open-source LLM, we first sample an original step and transform it into a sneaky step using the sneaky generator. We subsequently use the same LLM to complete the entire reasoning process after the original/sneaky steps, and this is repeated $N$ times. If the original step achieves a relatively high success rate while the sneaky step results in a significantly lower success rate, we consider this pair of steps to represent correct and incorrect steps, respectively. In our experiment, we adopt a strict criterion to ensure data quality. If the original step achieves a success rate greater than or equal to 75%, while the sneaky step results in a success rate of 0%, we then collect this pair of steps for subsequent adversarial games.

## 3.3 Initializing Step Critic

Based on the results from ProcessBench [27], reasoning models such as QwQ [22] and distilled R1 models [21] outperform non-reasoning models such as GPT when serving as critic models. However, the lengthy reasoning process in R1 leads to slow and redundant model generation, and its instruction-following capability is relatively poor, often failing to produce a concise critique with a definite conclusion about the correctness of a step. We therefore combine the strengths of both types of models when initializing the critic.

Specifically, we prompt DeepSeek-R1-Distill-Qwen-7B as a critic, taking problem $p$, partial solutions $\tau_{:k-1}$, and mixed correct and incorrect steps $t_k$ from PRM800K dataset as inputs, to collect long critiques. We then employ GPT-4 to rewrite them into brief standardized critiques $Q_t$ (see Sec. B). Concretely, $Q_t$ contains an analysis of the partial solution and the current last step, as well as a definite conclusion regarding the correctness of the step. This also simplifies the task and facilitates the use of SFT for policy initialization. Additionally, when preparing the training data for the critic, we mix the steps labeled as correct and incorrect in PRM800K at a 1:1 ratio to ensure the critic's capabilities are balanced. We utilize human annotations from PRM800K to filter around 21.8K correctly generated critiques $\overline{Q_t}$. Similarly, we prepare behavior cloning dataset $(\mathbf{x}, \mathbf{y}) \sim \mathcal{D}_{bc}^C$ for the critic, where $\mathbf{x} = (p, \tau_{:k-1}, t_k)$ as input, and $\mathbf{y} = \overline{Q_t}$ as output. We then finetune the base model using SFT loss (1) to obtain an initial policy $C_0$ for the critic.

## 3.4 Adversarial Game

We further reinforce the models' correct behavior and continuously improve their performance, avoiding the limitations related to the scale and distribution of human-annotated PRM800K. Inspired by recent self-play practices [19, 47], we propose a step-level adversarial game between the sneaky generator and step critic, enabling continuous reward generation and self-evolution of the two roles.

In each iteration of the adversarial game, we begin by using LLM solvers to generate a set of original step-by-step solutions for each problem. To ensure data diversity, we employ various LLM solvers from different model families, with sizes ranging from 7B to 32B, thereby enriching the diversity of sample styles. We then design an adversarial game for the two roles based on these solutions. Single steps are randomly selected from solutions for sneaky transformation, and the incorrect steps successfully produced by the sneaky generator are then fed into the critic to generate critiques. In addition to ensuring that the generated step contains an error, we expect the sneaky generator to generate incorrect steps with subtle flaws that can fool and challenge the critic. Meanwhile, the critic should be powerful enough to avoid being misled by any errors and provide an accurate critique.

In this game, we can set the rewards for the sneaky generator and the critic respectively in an adversarial instance as follows:

$$R_{sneaky} = \begin{cases} 1, & \text{Sneaky Generator Wins} \\ -1, & \text{Sneaky Generator Loses} \end{cases} \tag{2}$$

$$R_{critic} = \begin{cases} 1, & \text{Critic Wins} \\ -1, & \text{Critic Loses} \end{cases} \tag{3}$$

This opposing optimization goal enables both the sneaky generator and the critic to continuously improve their performance, achieving iterative self-evolution.

## 3.5 Evolving via Reinforcement Learning

In each iteration, after obtaining positive and negative samples through the adversarial games, we apply offline reinforcement learning to the critic and sneaky generator, respectively, enabling self-improvement of both roles based on the game result. Specifically, we adopt the following optimization objective to achieve stable RL training:

$$\nabla_\theta \hat{\mathcal{L}}(\theta) = \mathbb{E}_{\mathbf{x} \sim \mathcal{D}, \mathbf{y} \sim \pi_{\text{old}}(\mathbf{y}|\mathbf{x})} \left[ \frac{\pi_\theta(\mathbf{y}|\mathbf{x})}{\pi_{\text{old}}(\mathbf{y}|\mathbf{x})} \cdot \hat{A}^{\pi_{\text{old}}}(\mathbf{x}, \mathbf{y}) \cdot \nabla_\theta \log \pi_\theta(\mathbf{y}|\mathbf{x}) \right], \tag{4}$$

where $\pi_{\text{old}}$ denotes the policy used to collect the offline dataset, $\frac{\pi_\theta(\mathbf{y}|\mathbf{x})}{\pi_{\text{old}}(\mathbf{y}|\mathbf{x})}$ is the importance ratio, and $\hat{A}^{\pi_{\text{old}}}$ represents the advantage estimation. Inspired by recent RLOO [52] and GRPO [53], we formulate $\hat{A}^{\pi_{\text{old}}} = R(\mathbf{x}, \mathbf{y}) - b - \beta \text{KL}[\pi_\theta \| \pi_{\text{ref}}]$, where a baseline $b$ (the average reward of all samples) is subtracted for advantage estimation, and a Kullback-Leibler (KL) penalty is added to regularize the policy $\pi_\theta$ and prevent it from deviating too far from the initial policy $\pi_{\text{ref}}$.

For the sneaky generator, considering that we also need it to generate actual incorrect steps, we treat sneaky steps that fail to affect the problem-solving success rate as negative samples. Additionally, sneaky steps that successfully impact the LLM success rate but do not deceive the critic will also be considered negative samples. Meanwhile, the ones that can both influence the LLM success rate and deceive the critic are considered positive samples. Consequently, our data for training the sneaky generator includes a 1:1:1 ratio of positive samples and two types of negative samples.

As for the critic, we mix some correct steps from correct solutions with some incorrect steps generated by the sneaky generator for the critic to predict. The samples that the critic successfully predicts receive a positive reward, while those that are incorrectly predicted receive a negative reward. Ultimately, positive/negative samples each constitute half of the total samples.

Based on the adversarial game, we apply iterative training to enable continuous evolution of the two roles. Specifically, in each iteration, the newly updated policies re-engage in the adversarial game to generate new data for training, thereby evolving themselves further. Additionally, we observe an interesting phenomenon that more balanced adversarial games contribute to the self-evolution of models. In fact, the initial sneaky generator $S_0$ is weaker than the initial critic $C_0$, resulting in an unbalanced win rate. Moreover, $S_1$ obtained through synchronous iteration is even weaker than $C_1$. Therefore, we adopt an asymmetric evolution strategy, where $S_1$ competes against $C_0$ in a more balanced game to generate the second round of data. This enables $C_2$ trained in the second round to further improve its performance. Such a strategy is analogous to humans preferring to improve their skills in chess by playing against equally matched opponents. We provide more detailed analyses of the evolving strategies in Sec. 4.3.

## 3.6 Enhancing LLM Reasoning

Previous process reward models (PRMs) [26, 27] require scoring each step of the fully generated solutions and then integrate all the scores. However, after the first reasoning step error occurs, the LLM should promptly correct the mistake. Continuing to generate more potentially flawed reasoning steps after an erroneous step is unnecessary, and the scores produced are unreliable. In contrast, we propose a new approach that directly employs a critic to assist the LLM in searching for reasoning steps. During testing, we use '\n\n' to control the LLM to output one step at a time, allowing the critic to verify the correctness of each step. If the step is correct, the search continues; if incorrect, the LLM is required to regenerate the step (up to five attempts before skipping). Our SPC effectively enhances the reasoning performance of the LLM using this approach.

# 4 Experiments

## 4.1 Experimental Settings

**Evaluation** We adopt PRM800K [23], ProcessBench [27], and DeltaBench [35] that include human annotations of mathematical reasoning steps for evaluation. The original setting of ProcessBench and DeltaBench is to identify the position of the first or all errors in a complete solution. We argue that, in practical scenarios, a critic can enhance reasoning performance by identifying the incorrect

Table 1: Comparison of recall on ProcessBench. We evaluate different models on their ability to assess the correctness of the current step, instead of only predicting the index of the first error in the complete solution. 'Round 0' refers to the initialized critic model.

| Models | GSM8K | MATH | Olympiad-Bench | Omni-MATH | Average |
|---|---|---|---|---|---|
| *Process Reward Models (PRMs)* | | | | | |
| Math-Shepherd-PRM-7B [26] | 58.0 | 58.4 | 68.0 | 64.1 | 62.1 |
| Qwen2.5-Math-7B-PRM800K [27] | 77.0 | 72.9 | 66.9 | 62.1 | 69.7 |
| *Prompting LLMs as Critic Models* | | | | | |
| Llama-3.1-8B-Instruct [10] | 59.5 | 57.7 | 53.6 | 53.9 | 56.2 |
| Llama-3.1-70B-Instruct [10] | 67.2 | 62.8 | 61.7 | 61.9 | 63.4 |
| Qwen2.5-7B-Instruct [12] | 64.2 | 64.0 | 62.1 | 60.8 | 62.8 |
| Qwen2.5-32B-Instruct [12] | 76.2 | 68.1 | 68.9 | 63.9 | 69.3 |
| GPT-4o [6] | 75.5 | 70.5 | 70.0 | 64.5 | 70.1 |
| DeepSeek-R1-Distill-Qwen-7B [21] | 79.0 | **81.3** | 73.4 | 67.3 | 75.2 |
| *Our Critic Models* | | | | | |
| SPC (Round 0) | 78.0 | 74.1 | 67.8 | 63.2 | 70.8 |
| SPC (Round 1) | 82.0 | 80.3 | 74.8 | **70.3** | 76.8 |
| SPC (Round 2) | **84.2** | 80.8 | **76.5** | 69.2 | **77.7** |

step and requiring the LLM to regenerate, with no need to wait for completing all the steps. We therefore extract a 1:1 ratio of correct and erroneous steps from each benchmark, only retain the reasoning process before these steps as a partial solution, and discard the reasoning steps after these steps. Besides, we evaluate the effectiveness of the critic models in assisting LLMs to solve math problems on MATH500 [54] and AIME2024 [37]. More evaluation details are provided in Sec. C.

**Baselines** Following ProcessBench [27], we primarily evaluate two types of baselines, namely Process Reward Models (PRMs) and prompting LLMs as critic models. For PRMs, we select two representative methods, namely Math-Shepherd [26] and Qwen2.5-Math-7B-PRM800K [27]. Math-Shepherd trains a process reward model through an automated data annotation process and can be utilized to rank multiple outputs or ensemble them to enhance reasoning performance. Qwen2.5-Math-7B-PRM800K is based on the advanced math-specialized model Qwen2.5-Math-7B [55], and is further fine-tuned with the PRM800K dataset, obtaining state-of-the-art performance among PRMs. We also prompt multiple types of LLMs to serve as critic models, using the same prompts as our SPC. Several representative models, including Llama [10], Qwen [12], R1 [21], and GPT-4o [6], are selected as baselines.

### 4.2 Main Results

**Critic Performance on Reasoning Process Benchmarks** As shown in Tabs. 1 and 2, we compare our critic models with other baselines on 3 math-related reasoning process benchmarks to evaluate the abilities of predicting step correctness. We can observe that: (1) Our proposed SPC is gradually evolving and achieves state-of-the-art performance among all models. For example, the average performance on ProcessBench has improved from 70.8% to 77.7%, and on DeltaBench from 54.9% to 60.5%. (2) On all benchmarks, our method outperforms the latest PRMs specifically designed for scoring steps. (3) The performance of prompting LLMs as critics is not as good as SPC. Our method outperforms the distilled R1 model with the same size of 7B parameters. (4) Some baselines (PRMs and prompting Llama) have imbalanced recall between correct and error steps, leading to poor harmonic mean, whereas our critic is more balanced. (5) Our critic is trained on short CoT data (from Qwen and Llama), but it is able to generalize to long CoT reasoning steps (e.g., R1 [21] and QwQ [22]) in DeltaBench. In contrast, the two PRMs trained on short CoT show a significant performance decline in DeltaBench, with HarMean scores of only 14.3% and 41.3%, respectively.

**The Effectiveness of Guiding Test-Time Search** Existing PRMs can enhance performance by ranking the completely generated reasoning steps or by aggregating scores using self-consistency [2, 26]. We apply the proposed SPC to LLM reasoning search, utilizing SPC to check the correctness of each step and regenerating the step if it is incorrect (up to 5 retries). Moreover, SPC can be combined with self-consistency by conducting a majority vote over several independent searches. For a fair

Table 2: Comparison of our SPC with baselines on the test set of PRM800K [23] and DeltaBench [35], where human-annotated correct and erroneous steps are extracted to evaluate the recall of critiques. "Correct" and "Error" represent the recall on correct and erroneous steps, respectively. "Average" denotes their arithmetic average and "HarMean" refers to their harmonic mean.

| Models | PRM800K | | | | DeltaBench | | | |
|---|---|---|---|---|---|---|---|---|
| | Average | HarMean | Correct | Error | Average | HarMean | Correct | Error |
| *Process Reward Models (PRMs)* | | | | | | | | |
| Math-Shepherd-PRM-7B [26] | 50.0 | 49.5 | 55.2 | 44.8 | 53.3 | 14.3 | 7.69 | **98.8** |
| Qwen2.5-Math-7B-PRM800K [27] | 73.6 | 73.6 | 74.4 | 72.8 | 58.5 | 41.3 | **90.1** | 26.8 |
| *Prompting LLMs as Critic Models* | | | | | | | | |
| Llama-3.1-8B-Instruct [10] | 51.9 | 30.5 | 18.6 | 85.2 | 49.1 | 6.38 | 3.30 | 95.0 |
| Llama-3.1-70B-Instruct [10] | 54.6 | 38.9 | 25.3 | 83.9 | 44.6 | 20.3 | 11.7 | 77.5 |
| Qwen2.5-7B-instruct [12] | 52.8 | 37.2 | 24.1 | 81.6 | 48.2 | 33.8 | 21.8 | 74.7 |
| Qwen2.5-32B-instruct [12] | 59.0 | 50.5 | 36.6 | 81.4 | 44.7 | 33.0 | 21.8 | 67.6 |
| GPT-4o [6] | 68.5 | 68.4 | 70.3 | 66.6 | 49.9 | 48.7 | 42.0 | 57.9 |
| DeepSeek-R1-Distill-Qwen-7B [21] | 71.4 | 71.2 | 67.3 | 75.5 | 50.9 | 50.6 | 54.9 | 46.9 |
| *Our Critic Models* | | | | | | | | |
| SPC (Round 0) | 71.0 | 70.8 | 67.8 | 74.2 | 54.9 | 53.5 | 45.9 | 64.0 |
| SPC (Round 1) | 72.8 | 70.3 | 59.4 | **86.1** | 58.8 | 57.3 | 68.4 | 49.3 |
| SPC (Round 2) | **75.8** | **75.8** | 74.8 | 76.9 | **60.5** | **59.5** | 68.2 | 52.8 |

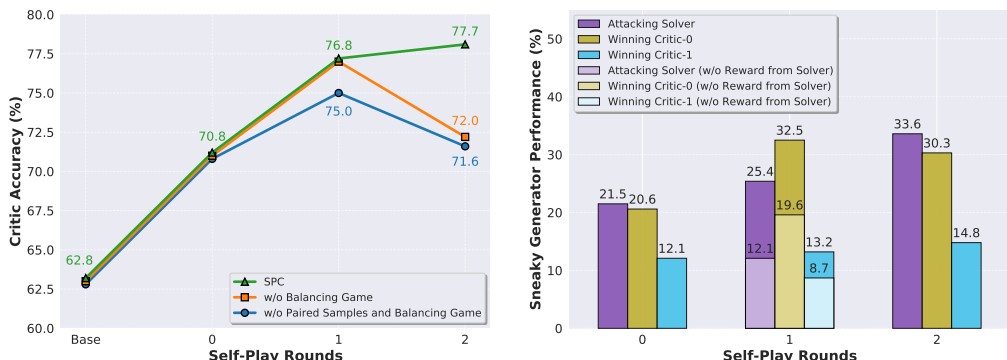

Figure 3: Ablation study of our critic and sneaky generator. Left: The impact of different strategies on evolving critic models. Right: The success rate of sneaky generator in attacking LLM solver and its win rate against round 0 and round 1 critics.

comparison, all methods incorporating self-consistency sample 5 outputs in our experiments. In addition, for experiments without using self-consistency, we run them at least three times and average the results to reduce randomness. As shown in Tab. 3, on two popular benchmarks MATH500 [54] and AIME2024 [37], SPC significantly improves the performance of three types of LLM solvers, and outperforms five baseline verifiers. For instance, using the Qwen Solver at AIME2024, our SPC combined with Self-Consistency achieves a problem-solving accuracy of 23.3%, which is superior to the 16.7% accuracy of Self-Consistency + Qwen2.5-Math-7B-PRM800K. Notably, our SPC is trained using only short CoT data, yet it can still generalize to the DeepSeek-R1-Distill-Qwen-7B model, which outputs in a long CoT style. It achieves 94.0% accuracy on MATH500, whereas Math-Shepherd and Qwen2.5-Math-7B-PRM800K achieve only 89.2% and 91.8%, respectively.

## 4.3 Ablation Study

**The Impact of Different Strategies on Evolving Critic** In Fig. 3 (left), we test critic models on ProcessBench, demonstrating the impact of different adversarial training methods. We refer to the sneaky generator and critic initialized after SFT as Sneaky-0 and Critic-0, respectively, while Sneaky-$n$ and Critic-$n$ represent models trained with $n$ rounds of self-play adversarial data. In round 1, Sneaky-1 and Critic-1 are trained using data generated from the adversarial game between

Table 3: Performance of various methods for assisting different LLMs in math reasoning. By integrating Self-Consistency with our SPC, we achieve the best results across three types of LLMs on MATH500 and AIME2024 datasets.

| Solvers | Verifiers | MATH500 | AIME2024 |
|---|---|---|---|
| Llama-3.1-8B-Instruct [10] | w/o | 47.0 | 4.27 |
| | Self-Consistency [2] | 55.6 | 3.33 |
| | Math-Shepherd [26] | 52.4 | 3.33 |
| | Qwen2.5-Math-7B-PRM800K [27] | 54.6 | 3.33 |
| | Self-Consistency + Math-Shepherd | 53.6 | 6.67 |
| | Self-Consistency + Qwen2.5-Math-7B-PRM800K | 60.4 | 3.33 |
| | SPC (Ours) | 54.5 | 5.63 |
| | Self-Consistency + SPC (Ours) | **62.8** | **6.67** |
| Qwen2.5-32B-Instruct [12] | w/o | 78.0 | 14.4 |
| | Self-Consistency | 82.0 | 16.7 |
| | Math-Shepherd | 78.8 | 13.3 |
| | Qwen2.5-Math-7B-PRM800K | 82.8 | 16.7 |
| | Self-Consistency + Math-Shepherd | 80.8 | 13.3 |
| | Self-Consistency + Qwen2.5-Math-7B-PRM800K | 84.6 | 16.7 |
| | SPC (Ours) | 83.0 | 17.7 |
| | Self-Consistency + SPC (Ours) | **85.2** | **23.3** |
| DeepSeek-R1-Distill-Qwen-7B [21] | w/o | 87.7 | 53.8 |
| | Self-Consistency | 92.2 | 70.0 |
| | Math-Shepherd | 87.0 | 53.3 |
| | Qwen2.5-Math-7B-PRM800K | 84.2 | 63.3 |
| | Self-Consistency + Math-Shepherd | 89.2 | 60.0 |
| | Self-Consistency + Qwen2.5-Math-7B-PRM800K | 91.8 | 73.3 |
| | SPC (Ours) | 92.3 | 52.6 |
| | Self-Consistency + SPC (Ours) | **94.0** | **73.3** |

**Sneaky-0 and Critic-0.** For each successfully transformed erroneous step, we have the critic predict four critiques, which may include both correct and incorrect predictions, forming a pair of positive and negative samples with the same input but different outputs. **This method of constructing paired samples is more effective in RL training**, improving the critic from 70.8% in round 0 to 76.8%, whereas not constructing paired samples only achieves a performance of 75.0%.

For round 2, we explore two evolving strategies. (1) Generating round 2 data using the confrontation between Sneaky-1 and Critic-1 and mixing it with the data from round 1. We observe a significant performance decline in the critic trained with this setting, dropping from 76.8% to 72.0%, possibly due to overfitting. We notice that the win rate of Sneaky-1 against Critic-1 is only 13.2%. Therefore, such an overly unbalanced game might prevent the critic from learning new knowledge from the adversarial process, similar to how humans need opponents of comparable skill levels when playing chess. Therefore, we adopt another setting: (2) Generating data through the game between Sneaky-1 and Critic-0, given that Sneaky-1 had a win rate of 32.5% against Critic-0. We then mix the data from both rounds for training Critic-0 and update it as Critic-2. **Balancing the game prevents performance degradation and enables self-evolution**, improving SPC's performance to 77.7%.

**The Performance of Sneaky Generator** As shown in Fig. 3 (right), we analyze sneaky generators' success rates in attacking Qwen-2.5-7B-Instruct solver, as well as their win rates against Critic-0 and Critic-1. It is observed that the proportion of successful attacks on the solver gradually increases from 21.5% to 33.6%, as the sneaky generator iterates. We then feed successfully generated erroneous steps to the critic models. Sneaky generators' win rates against Critic-0 increase from 20.6% (Sneaky-0) to 30.3% (Sneaky-2). **Overall, the performance of the sneaky generators is iteratively improved.**

We also analyze a training setting without adding failed attacks on the solver as negative samples, using only successfully generated erroneous steps to construct positive/negative samples for training Sneaky-1, referred to as "w/o Reward from Solver" with lighter colors. We find that this approach severely impacts the performance of the sneaky generator, significantly reducing the proportion of successful attacks to 12.1%. Among the successfully attacked samples, the proportion that could deceive the critic is also very low, achieving a 19.6% win rate against Critic-0. Therefore, **it is crucial to ensure that the sneaky generator receives rewards from both the solver and the critic.**

# 5 Conclusion

In this paper, we propose a self-play critic with the ability of detecting step-level LLMs reasoning errors. Specifically, we design a sneaky generator to produce incorrect steps and a critic to assess the correctness of each step. Through the adversarial game between these two models, we can continuously generate positive and negative samples for reinforcement learning. The results on three reasoning process evaluation benchmarks fully demonstrate the effectiveness of our SPC. Furthermore, we apply SPC to assist LLMs' test-time search, further enhancing their reasoning performance.

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

# Appendices

## A  Limitations and Societal Impact

Our SPC continuously generates data for reinforcement learning through adversarial games and has achieved impressive results. However, our current experiments are limited to several representative mathematical reasoning tasks. In future work, we plan to extend our approach to more general domains to further demonstrate the potential of the proposed framework.

Potential negative societal impacts of this work may include the misuse of a sneaky generator. For example, training a general sneaky generator to produce false and misleading information. On the other hand, enhancing the robustness of LLMs against attacks and training a general critic to automate the review of false information on the internet are also worthwhile research directions.

## B  Prompts

We demonstrate the prompts for generating data to initialize the sneaky generator and critic model.

Fig. 4 shows the prompts for querying GPT-4 (gpt-4-turbo-2024-04-09) to obtain sneaky transformations. These prompts include the five predefined error types, as well as correct/incorrect step pairs extracted from PRM800K. GPT-4 needs to first select a corresponding error type and then output the transformation process.

Figs. 5 and 6 illustrate the prompts we used to collect step-level critiques for initializing the critic model. We first use the prompts in Fig. 5 to feed data from PRM800K into DeepSeek-R1-Distill-Qwen-7B, collecting a batch of raw critiques. However, these critiques often do not follow our instructions in a standard format, making it difficult for us to assess the correctness of the critiques. Additionally, the responses are often too lengthy and include a lot of reflection and exploration, which is not conducive to performing SFT on the base model. Therefore, we feed these raw critiques (referred to as draft critiques in the prompt) into GPT-4o (gpt-4o-2024-08-06) for refinement, as shown in Fig. 6. By leveraging GPT-4o's strong instruction-following capabilities, we summarize the unstructured critiques into a concise and standardized version for subsequent SFT training.

When actually training the sneaky generator and critic model, we make slight modifications to the prompts mentioned above to avoid including incorrect steps that should be in the LLM output and unnecessary information, such as draft critiques. As shown in Figs. 7 and 8, we demonstrate the prompt templates for training the sneaky generator and critic model, respectively. These templates also remain unchanged during testing, data generation for self-play, and reinforcement learning processes.

## C  More Details

### C.1  Evaluation Details

PRM800K [23] is a dataset collected by OpenAI for training and evaluating process supervision models. It is large in scale, containing 800K GPT-generated reasoning steps with human-annotated correctness. Additionally, PRM800K includes many pairs of correct and incorrect steps that share a common partial solution. Therefore, we construct 1,341 pairs of steps from the test split to evaluate model performance.

ProcessBench [27] is a benchmark with human-annotated step correctness, but it only includes a test set for evaluating models. Compared to PRM800k, the reasoning steps in ProcessBench are more diverse, comprising 3,400 cases from 12 different LLMs. All of these are math problems sourced from four datasets, including GSM8K [56], MATH [54], OlympiadBench [57] and Omni-MATH [58]. For incorrect solutions, we only retain the first incorrect step, while we randomly sample one correct step from a correct solution. We then feed the mixed 1,700 correct steps and 1,700 incorrect steps along with their corresponding partial solutions into the critic models.

DeltaBench [35] is the newest process benchmark focusing on evaluating long CoT collected from different open-source reasoning models, such as R1 [21] and QwQ [22]. We only utilize the math-related problems in this benchmark to evaluate model performance. Similarly, we retain the

labeled erroneous steps and sample the same number of correct steps, totaling 1,542. Given that existing PRMs baselines and our adversarial data generation process only collect short CoT data, this benchmark is more challenging and can be utilized to evaluate the effectiveness of our critic models in generalizing to popular reasoning models with long CoTs.

MATH500 [23, 54] and AIME2024 [37] are two highly popular benchmarks used to assess the mathematical reasoning abilities of LLMs. The former consists of 500 competition-level math problems, while the latter is derived from the American Invitational Mathematics Examination 2024. We evaluate the performance of LLMs on these two benchmarks when assisted by different verifiers in reasoning.

## C.2 Preparing Data

For the SFT phase of the critic, we utilize the reasoning process data from PRM800K, along with prompting GPT-4 and DeepSeek-R1-Distill-Qwen-7B [21], to generate the required step-level critique data. We employ human annotations from PRM800 to filter out correctly generated data, ultimately obtaining 21.8K data, including 9.4K correct steps and 12.4K incorrect steps. As for the sneaky generator, we also prompt GPT-4 to teach the LLM to transform the correct steps from PRM800K into incorrect steps, finally collecting 13K data for SFT.

During the self-play phase, we use problems from the training set of PRM800K [23] to generate adversarial data for reinforcement learning. We use a total of three types of LLM solvers (Llama-3.1-8B-Instruct [10], Qwen2.5-7B-Instruct and Qwen2.5-32B-Instruct [12])) to provide the initial reasoning steps, in order to sample the correct steps and perform a sneaky transformation, which are then fed to the critic for adversarial self-play. Since we need the LLM to complete from both the original correct step and the sneaky step generated by the sneaky generator and compare the problem-solving success rate to determine whether the sneaky step contains an error that can truly affect the reasoning process, we first pre-generate 4 solutions for each problem and filter out those that have a success rate of 0, which are inherently unsolvable. For the remaining problems, we consider those with a success rate of 1/4 and 2/4 as medium difficulty level for this LLM, while those with a success rate of 3/4 and 4/4 are considered relatively easy. We primarily use medium-level problems to construct the training data, which ultimately accounts for 90% of the dataset, while easy problems are retained at about 10% because the model can already solve these problems smoothly without much additional learning and we only need a small amount of such data. These filtered medium-level problems will have 16 solutions generated by each LLM solver and easy problems will directly use the pregenerated 4 solutions. For each correctly predicted solution, one correct step is sampled and performed sneaky transformation. The successfully transformed incorrect steps are then further filtered for adversarial self-play.

The first round of self-play occurs between two SFT models (sneaky-0 and critic-0). We collect 6.4K data for the critic model for reinforcement learning, with a 1:1 ratio of positive to negative samples. Meanwhile, the sneaky generator receives 6K data, divided equally (2K each) into three scenarios: failing to attack the LLM solver, successfully attacking the LLM solver but losing to the critic, and successfully attacking the LLM solver while defeating the critic. The collected data in round 1 help us iteratively update the models to sneaky-1 and critic-1. As mentioned in Sec. 3.5, we balance the adversarial game to collect training data. Therefore, the second round of self-play occurs between sneaky-1 and critic-0. We further collect 6.8K data for the critic model, maintaining a 1:1 ratio between positive and negative samples, while continuing to gather 6K data for the sneaky generator, with the three scenarios still evenly distributed at 1/3 each. Finally, the data from two self-play rounds is merged to conduct offline reinforcement learning on sneaky-0 and critic-0, updating them to sneaky-2 and critic-2, respectively.

## C.3 Training Hyperparameters

In the SFT initialization phase for both sneaky generator and critic models, we employ a batch size of 64 and a learning rate of 5e-6. We train the models for 3 epochs, with the maximum sequence length set to 4,096. To ensure both stability and convergence during training, we also incorporate a KL penalty into the training loss, setting the KL coefficient at 0.1. During the reinforcement learning of the self-play phase, we keep the batch size as 64 but use a learning rate of 2e-6. Except for setting

the KL coefficient at 0.1, we also add an SFT loss with a coefficient of 0.15 to ensure the stability of RL training.

# D   Case Analysis

As shown in Fig. 9, we present a comparison of critiques provided by two SPCs trained in round 0 and round 2. The input to the SPCs includes not only the system prompt (Sec. B but also the problem, partial solution, and the last step of the current reasoning process within the blue box in the figure. The last step contains a logical error, mistakenly identifying two expressions as unmatched. The critique from the round 0 SPC considers the last step to be correct, agreeing with the view that the expressions are inconsistent. However, the round 2 SPC, evolved through self-play training, accurately identifies the type of error, namely a logical error (underlined in the figure), recognizing that the two sides of the equation can be equivalently substituted. The entire analysis process is clear and coherent, ultimately leading to the correct prediction that this step is incorrect.

## User Prompt

You are a math teacher. Given a math problem and a partial solution, you need to convert the last step of this partial solution to an incorrect step, while ensuring the incorrect step is subtle enough to be easily overlooked.

You should follow these steps to converse the last step:
1. Analyze the given partial solution and the last step. Clearly explain the solving process in the last step.
2. Choose an appropriate error type from the "Predefined Error Types" to complete the error generation. Specify your error generation method based on the current case, making the error less noticeable.
3. Step-by-step, write out the detailed error generation process for converting the Correct Last Step into the Incorrect Step.
4. Wrap the final incorrect step with <Answer> </Answer>.

Note: For each question, you will be given a reference incorrect last step. You need to convert to this reference incorrect last step, but **you must not reveal that you know this reference step in advance during the conversion process!**

# Predefined Error Types
Error Type 1: Logical Error
Reference cases:
- incorrect orientation of geometric figures
- systematic counting error
- incomplete and inaccurate listing of factors
- Incorrect interpretation and connection of definitions
Error Type 2: calculation error
Reference cases:
- make incorrect simplification
- make incorrect factorization
- invalid algebraic operation
- substitution error
Error Type 3: Misunderstanding the Conditions.
Reference cases:
- Use imcomplete Condition
- Use contradictory condition
- misinterpretation of problem requirements
Error Type 4: Use Incorrect Rules/Formulas/Properties
Reference cases:
- incorrect identification of prime numbers
- misapplication of properties of complex numbers
- introduction of irrelevant inequality
Error Type 5: Incorrect Approach
Reference cases:
- incomplete solution

# Your Task

## Problem
{problem}

## Partial Solution
{partial_solution}

## Correct Next Step
{correct_step}

## (Reference) Incorrect Next Step
{incorrect_step}

## Your response

Figure 4: Prompt for querying GPT-4 to collect raw data of sneaky transformation CoT.

---

**User Prompt**

You are a helpful critic. Given a math Problem, a Partial Solution, the current Last Step of the solution, You need to provide a critique for the correctness of the Last Step.

You need to response a step-by-step analysis:
1. Analyzing the general thought of the Partial Solution.
2. Critique. You should write a brief critique here. This part should also maintain logical coherence with the summary of the general thought of the Partial Solution.
3. Conclusion. At the end of the response, output \\boxed{{Correct}} or \\boxed{{Incorrect}} to represent the correctness of the Last Step.

## Problem
{problem}

## Partial Solution
{partial_solution}

## Last Step
{last_step}

---

Figure 5: Prompt for querying DeepSeek-R1-Distill-Qwen-7B to collect raw critiques with long CoT.

---

**System Prompt**

You are a helpful critic. Given a math Problem, a Partial Solution, the current Last Step of the solution, You need to provide a critique for the correctness of the Last Step. You already have a Draft Critique, so you only need to rewrite it in a clearer and more concise format.

You need to response a step-by-step analysis:
1. Analyzing the general thought of the Partial Solution. If the Draft Critique includes this analysis, you can directly summarize from it.
2. Critique. You should write a new version of brief critique here. The current Draft Critique is accurate but may contain redundant information, such as extensive consideration and attempts at derivation and analysis of the problem. You only need to select the useful analysis for the Last Step from it. This part should also maintain logical coherence with the summary of the general thought of the Partial Solution.
3. Conclusion. Please draw a conclusion about the correctness of the Last Step. Based on the analysis of the provided Draft Critique, determine whether the Last Step is correct or incorrect. If it is incorrect, you should also summarize a specific type of error, such as calculation error, logical error, etc. At the end of the response, output <Answer>Correct</Answer> or <Answer>Incorrect</Answer> to represent the correctness of the Last Step.

NOTE:
1. You need to refer to the draft critique, but pretend you didn't know this information beforehand, avoiding phrases like "the critique". Just write a new version of the critique for the Problem, Partial Solution, and Last Step.
2. In your revised version of Critique, you only need to focus on the Last Step, and it is not necessary to solve the problem to obtain the final answer.
3. DO NOT write the conclusion first and then the explanations for it. Instead, in the Critique, you start with an analysis of the Last Step. Then, in the Conclusion, drawing a conclusion about whether the Last Step is correct or incorrect.
4. The current partial solution may be incomplete, which is not an error type. You only need to focus on the correctness of the Last Step.

**User Prompt**

## Problem
{problem}

## Partial Solution
{partial_solution}

## Last Step
{last_step}

## Draft Critique
{draft_critique}

---

Figure 6: Prompt for querying GPT-4o to rewrite a long critique into a brief and standardized critique.

Figure 7: Prompt for training the sneaky generator.

## System Prompt

You are a helpful critic. Given a math Problem, a Partial Solution, the current Last Step of the solution, You need to provide a critique for the correctness of the Last Step.

You need to response a step-by-step analysis:
1. Analyzing the general thought of the Partial Solution.
2. Critique. You should write a brief critique here. This part should also maintain logical coherence with the summary of the general thought of the Partial Solution.
3. Conclusion. At the end of the response, output <Answer>Correct</Answer> or <Answer>Incorrect</Answer> to represent the correctness of the Last Step.

## User Prompt

## Problem
{problem}

## Partial Solution
{partial_solution}

## Last Step
{last_step}

Figure 8: Prompt for training the critic model.

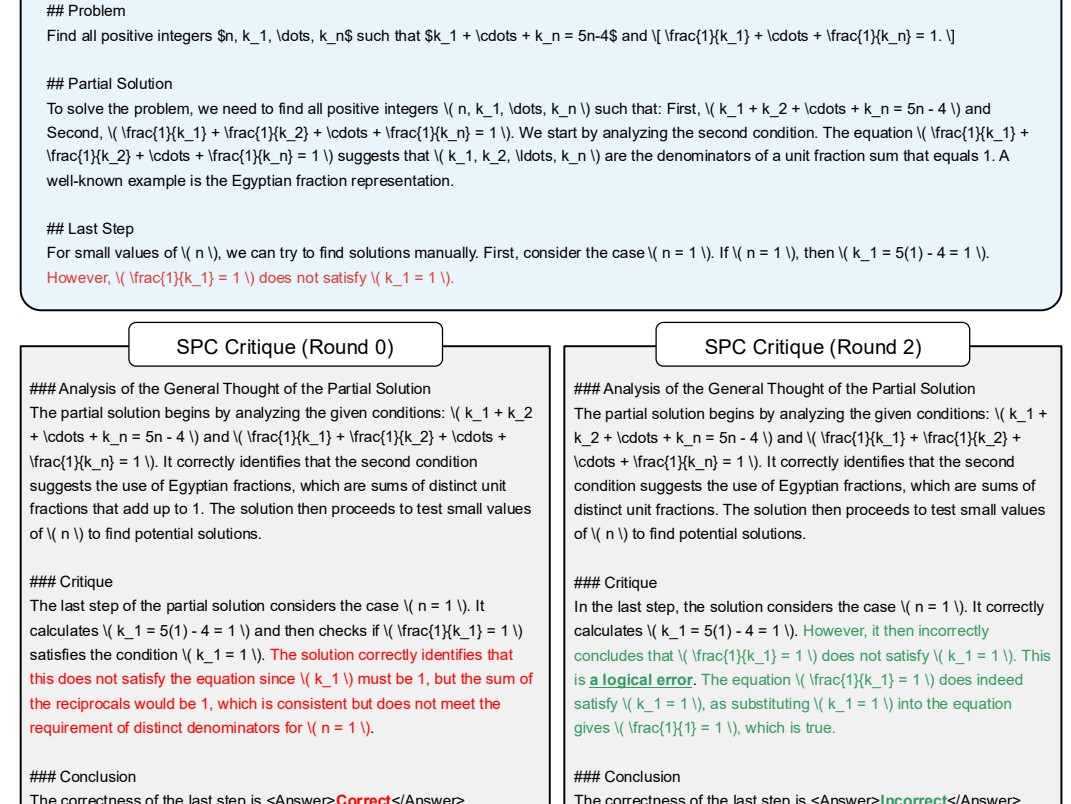

Figure 9: SPC critiques on ProcessBench before and after self-play training.

