# OpenReview forum: "SPC: Evolving Self-Play Critic via Adversarial Games for LLM Reasoning"
_NeurIPS.cc/2025/Conference — NeurIPS 2025 poster_

### Official Review · Reviewer_oHJP · 2025-06-11

**Clarity:** 4
**Significance:** 3
**Originality:** 3
**Rating:** 5
**Confidence:** 5

**Summary:**

This paper proposes a method for training a step-wise verifier/critic model for math reasoning CoTs. The method, Self-Play Critic (SPC), is trained via an adversarial 2-player game, where a generator is trained to introduce errors into reasoning while a critic is optimized to find the errors. Both the generator and critic are initialized with SFT policies using data created from PRM800k, and then trained via RL, with the critic receiving a reward for detecting a mistake and the generator receiving a reward for fooling the critic.

The method is used to train a Qwen-2.5-7B model, which is evaluated intrinsically on three PRM benchmarks (ProcessBench, PRM800k, and DeltaBench) where it shows improvements over both trained PRMs and advanced LLMs prompted to act as critic models. The method is evaluated extrinsically on 2 math reasoning tasks (MATH500 and AIME) where it is used in an overgenerate-and-filter approach (optionally combined with self-consistency). On the intrinsic evaluation, the method shows robust improvements over existing methods, while the size of the gains on the downstream evaluation is somewhat smaller. The ablations test matching generators with critics from earlier rounds, as matching from the same round makes the task too easy for the critic.

**Questions:**

1. Does SPC training generalize to other models besides Qwen?
2. Does SPC-trained Qwen benefit other models downstream?
3. What explains the relatively small performance improvements on the dowstream eval?

**Ethical Concerns:**

["NO or VERY MINOR ethics concerns only"]

**Final Justification:**

After reading the author response, I am raising my score, as most of the points raised above have been addressed. While I see the some concerns raised by other reviewers (limited to math, requiring startup data) as valid limitations, I think that these are shared by many other papers and this paper represents a step towards an adversarial data generation pipeline (while maybe falling short of showing that the solution works on a comprehensive set of diverse tasks).

**Limitations:**

Yes

**Quality:**

3

**Strengths And Weaknesses:**

**Strengths:**
- **Intuitive and useful idea**: The framing of step-wise verification as an adversarial task makes intuitive sense, and showing how these two models can be optimized via RL successfully is a useful contribution.
- **Outperforms PRMs**: on intrinsic evals of PRMs, the model trained via SPC outperforms several baselines, including proprietary models being prompted as critics and specialized PRMs.
- **Sensible analysis and useful details**: I appreciate that the paper shows improvements across rounds and has a clear analysis of what does (and does not) work in terms of pairing the sneaky generator with the critic at the same round vs. the prior round's critic.
- **Clear writing/method description**: The paper is easy to follow and well-written.

**Weaknesses**:
- **Generalization ability is unclear**: There are a couple questions I have about generalization which are left unanswered:
1. Does the method generalize to other models? The paper only trains one model (Qwen-2.5-7B-Instruct) and so it's not clear to me this would generalize to other models. It would be helpful to see results on other similar models to know that the method is not overfitting to one model.
2. Can a critic of one model type help other models at test time? As far as I can tell, the results in Table 3 are using Qwen both as the critic (trained with SPC) and as the reasoning model. While I understand that data was mixed from different models for training, it would be nice to see transfer here to other models. This wouldn't require any training, since you could use the SPC-trained Qwen model to do reranking on Llama or another model.
- **Extrinsic results are underwhelming**: The size of the gains in Table 3 are quite small, and in some cases using SPC without SC has lower performance than PRM800k trained Qwen (e.g. on AIME2024 with DS-R1-Distill-Qwen, though that may be due to small sample size in AIME). Further explanation of why the gains here are not as substantial would be helpful.
- **Further uses of SPC could be explored**: The main downstream use explored here is for a resampling approach, but PRMs can be used in more advanced ways, e.g. for refinement (https://arxiv.org/abs/2409.12147).

**Other comments**
Given the adversarial training nature of the method, I would have liked to see some more discussion on GANs

---

> ### Author Rebuttal · Authors · 2025-07-31
>
> We appreciate that you found our ideas intuitive and useful, achieved better results than PRMs, provided sensible analysis and useful details, and offered clear writing and method descriptions. We have also addressed your questions below.
>
> **W1. Generalization ability**
>
> For the critic model, we chose to train on Qwen2.5-7B-Instruct because it is one of the most popular and representative models, which allows us to quickly verify the effectiveness of our proposed SPC. In addition, many of our baselines are also 7B models, so using 7B models for training makes the comparison fairer. We will try to include results on additional models to further demonstrate the generality of our approach.
>
> You also suggest performing experiments where the critic model is used to assist other types of models during testing. We would like to clarify that this is **exactly what we did in the experiments shown in Table 3**. In the leftmost column of the table, we are referring to using the SPC model to assist different solvers in problem-solving, rather than using these different solvers to collect data. Specifically, we used three solvers for experiments: Llama-3.1-8B-Instruct, Qwen2.5-32B-Instruct, and DeepSeek-R1-Distill-Qwen-7B. These results demonstrate the generalization ability of the proposed SPC, as it can help different types of solvers further improve their performance.
>
> **W2. The performance of SPC in assisting different LLMs**
>
> The effect of our SPC in assisting DeepSeek-R1-Distill-Qwen-7B on the AIME2024 dataset is indeed not very significant. As you mentioned, this may be due to the small sample size in AIME, which introduces some randomness. Another possible reason is that AIME2024 is quite challenging, and a performance of 73.3% is already very high, making further substantial improvements difficult. We believe that conducting adversarial training on some highly challenging datasets may further enhance the effectiveness of SPC on AIME2024, which can be explored in the future.
>
> **W3. Further uses of SPC**
>
> We thank the reviewer for this insightful suggestion. We agree that the utility of SPC extends beyond resampling, such as refinement. To explore, we conducted a simple experiment. When our SPC critic identifies an incorrect reasoning step, we provide the solver with the previous erroneous step and the critic's feedback, guiding it to perform a targeted refinement (up to five attempts). Interestingly, we found that the refinement performance of Llama-3.1-8B-Instruct on MATH500 reached 63.4%, which is slightly higher than the 62.8% achieved with re-generation. We hypothesize that the moderate gain is due to the inherent difficulty of the solver performing the refinement task itself; the solver must recognize and overcome its initial flawed reasoning, which can "anchor" its subsequent attempts. This aligns with recent studies on the challenges of self-correction [1,2].
>
> Crucially, this inspires a compelling future direction to employ our SPC method in improving models' refinement ability. This would involve adapting the data sampling process to a refinement process during RL data collection. We believe this could be a very interesting direction for future work.
>
> [1] Kumar et al. "Training Language Models to Self-Correct via Reinforcement Learning." ICLR 2025.
>
> [2] Kamoi, Ryo, et al. "When can LLMs actually correct their own mistakes? a critical survey of self-correction of llms." TACL 2024.
>
> **Other comments - Discussion on GANs**
>
> Some early GANs [3, 4] achieve great success in image generation through adversarial training, but it is difficult to transfer this approach to NLP tasks. This is because images are continuous signals, which makes it easier for GANs to optimize the generator, allowing the generated images to gradually approach real images by minimizing the cross-entropy loss on the discriminator’s real/fake logits. However, for discrete tokens in text tasks, the optimization methods used by GANs are difficult to be effective.
>
> Our SPC is fundamentally different from the adversarial training framework of GANs on vision tasks. For text tasks with discrete tokens, we leverage reinforcement learning to achieve adversarial learning. Here, the sneaky generator’s goal in RL is not only to attack the critic, but also to generate more subtle errors. We finally obtain a step-level critic for assessing LLM reasoning and providing valuable feedback. This is fundamentally different from the mutual improvement framework of adversarial training in GANs.
>
> [3] Goodfellow et al. "Generative Adversarial Nets." NeurIPS 2014.
>
> [4] Karras et al. "A Style-Based Generator Architecture for Generative Adversarial Networks." CVPR 2019.

---

> > ### Comment · Reviewer_oHJP · 2025-08-03
> > **Response to updated results and discussion**
> >
> > Thanks for the response to my review -- the additional results on SPC training for Llama-3.1 are appreciated, as is the analysis on refinement (which I hope can be included in the paper). For the discussion on GANs, I had meant this more in the sense that there are fundamental thematic similarities between the approaches, but I also appreciate the differences the authors point out.
> >
> > Based on the additional results and clarifications, I am raising my score to 5 and recommending the paper be accepted.

---

> > > ### Author Response · Authors · 2025-08-04
> > >
> > > Thank you very much for your positive feedback and for raising your score to 5. We greatly appreciate your thoughtful comments and suggestions, which have helped us improve our work. We will include the refinement analysis in our revised version, and we will also add a discussion on the similarities and differences with GANs. Thank you again for your efforts in reviewing our work!

---

> ### Author Response · Authors · 2025-08-03
> **SPC training on Llama-3.1-8B-Instruct**
>
> To demonstrate the generality of our proposed SPC during training across different base models, we conducted some additional experiments on **Llama-3.1-8B-Instruct**, as shown in the table below. Specifically, we used the same data to first cold-start a critic based on Llama-3.1-8B-Instruct, and then collected data through adversarial self-play with a sneaky generator for subsequent RL training. Due to time constraints, we conducted only one round of adversarial training. The experimental results show that our Llama-based critic also achieves significant performance improvements across three benchmarks.
>
> | Methods       | ProcessBench | PRM800K | DeltaBench |
> |---------------|:--------------:|:---------:|:------------:|
> | Prompting     | 56.2         | 51.9    | 49.1       |
> | SPC (Round 0) | 61.1         | 58.9    | 55.8       |
> | SPC (Round 1) | 66.2         | 62.0    | 58.4       |
>
> Table: "Prompting" refers to the performance of the original Llama-3.1-8B-Instruct, while "round 0" and "round 1" represent the performance of our SPC (based on Llama-3.1-8B-Instruct) after cold start and after one round of self-play, respectively.
>
> These results validate that SPC can adapt well to other model family, verifying the generality of our proposed algorithm. We believe that these new results can address your first question regarding concerns about generalization.

---

### Official Review · Reviewer_pEmR · 2025-06-26

**Clarity:** 4
**Significance:** 2
**Originality:** 1
**Rating:** 3
**Confidence:** 5

**Summary:**

This paper tackles the problem of judging every single step in an LLM’s chain-of-thought without human labors. The authors propose a self-play framework—aptly named Self-Play Critic (SPC)—in which one model learns to generate sneaky & believable mistakes, while its adversarial model learns to catch them.

Thery start with PRM800K’s human-labelled step pairs to give both “sneaky generator” and “critic” a supervised warm-up. Then the generator learns to rewrite correct steps into subtly wrong ones (five error archetypes) based on the previous steps, and only those sneaky steps that make a downstream solver fail hard (success rate drop from ≥ 75 % to 0 %) are kept. Both gold and sneaky candidate steps are fed to the critic, which obtains reward if it spots the flaw, and otherwise the sneaky generator scores. Each win/loss is a ±1 reward. They store every interaction and run offline RL with KL anchoring so neither model drifts too far from its SFT roots.

After two self-play rounds the critic’s step-level accuracy climbs from 70.8 % to 77.7 % on ProcessBench and posts similar gains on PRM800K and DeltaBench, beating existing process reward models. When plugged into test-time reasoning, SPC nudges three different solvers to new highs on MATH500 and AIME 2024—e.g. a 7 B distilled-R1 model hits 94 % on MATH500 when guided by SPC versus 91.8 % with the best baseline verifier.

Overall, the work shows that step-level feedback can bootstrap itself: once you have a seed set, two small models can keep generating better supervision for each other, all without further human labeling.

**Questions:**

1. Why insist on the “≥ 75 % vs 0 %” success-rate gap? Did other cut-offs (e.g. 60 %→10 %) hurt critic learning or just add noise?

2.  All filters and rewards hinge on the three chosen solvers. How does SPC behave if the solver is weaker/stronger or writes in a very different style?

**Ethical Concerns:**

["NO or VERY MINOR ethics concerns only"]

**Limitations:**

Yes.

**Quality:**

3

**Strengths And Weaknesses:**

S1. Balanced, asymmetric evolution strategy: Authors recognise and solve the “generator too weak” problem by pairing the new generator with the previous critic; keeps win-rate ~30 % and drives continual improvement. Provided experimental support for the importance of balancing, which is not covered in similar adversarial training papers.

S2. Fine-grained error taxonomy: Five concrete error families (logical, calculation, mis-reading conditions, mis-using rules, wrong approach) give the generator a structured search space and make the critic’s feedback interpretable.

---

W1. Experiments confined to math; it is not clear whether similar sneaky generation strategy would be effective for more ecologically vaild & environment-dependent tasks, such as software development, web agent, os operation.

W2. Reward shaping still heuristic (binary ±1); sensitivity analysis limited.

W3. The current generator-verifier mutual enhancement effect from adverarial training methodology has been long known for reasoning tasks even before the age of LLM, which significantly undermines the originality of the work. The most important takeaway from this paper is perhaps the sneaky data synthesis strategy -- but the error taxonomy limits the domain transferability.

---

> ### Author Rebuttal · Authors · 2025-07-31
>
> We are pleased that you praised our introduction of the evolution strategy, which is not covered in similar adversarial training papers, as well as our implementation of a fine-grained error taxonomy. Below, we have carefully addressed your questions.
>
> **W1. Evaluation Tasks**
>
> Thank you for your comment. We also discussed this issue in the Limitations section (see Appendix A). We acknowledge that our current experiments are limited to mathematical tasks, as this domain provides high-quality, step-level human annotations that allow us to effectively evaluate the performance of our trained critic. **We also believe that this framework could be highly valuable in other domains**, such as the agent scenarios you mentioned. For example, the outcomes of multi-agent interactions can also be used to generate data for reinforcement learning. A well-trained critic can effectively assess the agent’s actions at each step and guide the decision-making process. Constructing high-quality evaluation benchmarks and extending our methods in these domains can serve as our future work.
>
> **W2. Reward Shaping**
>
> Our framework focuses on the design of the RL task and environment, and the reward signal is solely determined by whether the LLM wins or loses the game. Therefore, we adopted a binary reward (+1 for the winner, -1 for the loser) to directly reflect the competitive nature of the game and to ensure a clear learning signal for both models. **This reward design was also used in the self-play of AlphaGo Zero [1], demonstrating its robustness.** Thank you for your comments, and we will provide additional explanations and clarifications in our revised version.
>
> [1] Silver, D., Schrittwieser, J., Simonyan, K., et al. "Mastering the game of Go without human knowledge." Nature 2017.
>
> **W3. The originality of SPC**
>
> To the best of our knowledge, related research in self-play or adversarial training methodology can be categorized as follows:
> 1. Training in board games. For example, AlphaGo Zero [1] continuously collects data through self-play in the natural Go environment with the help of MCTS, using both the action probability distribution and the overall win rate estimation as learning targets, and updates the network via reinforcement learning.
> 2. Self-play learning for LLMs through existing games to achieve capability transfer [2, 3]. For instance, SPAG [2] leverages a party game called Taboo, where the LLM is required to guess a target word based on dialogue information, thereby improving the LLM’s reasoning ability.
> 3. Test-time self-play among LLMs without any training [4, 5]. For example, rStar [4] predefines a set of actions and combines them with MCTS for reasoning trajectory search, thus achieving a mutual generation-discrimination process and improving mathematical reasoning performance. This is indeed a solver-critic self-play during the testing phase to improve performance, but it cannot enable LLMs to self-evolve through self-play.
> 4. Some early GANs [6, 7] achieve great success in image generation through adversarial training, but it is difficult to transfer this approach to NLP tasks. This is because images are continuous signals, which makes it easier for GANs to optimize the generator, allowing the generated images to gradually approach real images by minimizing the cross-entropy loss on the discriminator’s real/fake logits. However, for discrete tokens in text tasks, the optimization methods used by GANs are difficult to be effective.
>
> In contrast, the originality and contributions of SPC lie in:
> 1. Compared to AlphaGo Zero, SPC extends the self-play approach to the recently popular domain of LLM reasoning. The main innovation is the careful design of **an adversarial training process for LLM tasks that lack a native game environment**, enabling continuous data generation for reinforcement learning.
> 2. Compared to SPAG, in the design of adversarial games, we do not simply utilize existing board or party games to indirectly improve LLM capabilities. Instead, SPC **directly designs a step-level generation and verification game for the critic**, enabling direct training of the LLM’s critic ability.
> 3. Compared to rStar, we combine the self-play framework with reinforcement learning to improve the model’s capabilities, rather than prompting LLMs and only performing test-time computation. Our newly developed self-play training framework **enables the critic to self-evolve through self-play**.
> 4. Our SPC is also fundamentally different from the adversarial training framework of GANs on vision tasks. For text tasks with discrete tokens, we **leverage reinforcement learning to achieve adversarial learning**. Here, the sneaky generator’s goal in RL is not only to attack the critic, but also to generate more subtle errors. We finally obtain a step-level critic for assessing LLM reasoning and providing valuable feedback. This is fundamentally different from the mutual improvement framework of adversarial training in GANs.
>
> Thank you for your constructive comments. Clarifying the difference of SPC to previous related adversarial training methodology helps better highlight our contribution and improve the quality of the paper. We will add these discussion into the Related Work in the revised version.
>
> [1] Silver et al. "Mastering the game of Go without human knowledge." Nature 2017.
>
> [2] Cheng et al. "Self-playing Adversarial Language Game Enhances LLM Reasoning." NeurIPS 2024.
>
> [3] Wu et al. "Enhance Reasoning for Large Language Models in the Game Werewolf." arXiv 2024.
>
> [4] Qi et al. "Mutual Reasoning Makes Smaller LLMs Stronger Problem-Solvers." ICLR 2025.
>
> [5] Xi et al. "Enhancing LLM Reasoning via Critique Models with Test-Time and Training-Time Supervision." arXiv 2024.
>
> [6] Goodfellow et al. "Generative Adversarial Nets." NeurIPS 2014.
>
> [7] Karras et al. "A Style-Based Generator Architecture for Generative Adversarial Networks." CVPR 2019.
>
> **Q1. Success-rate gap**
>
> Yes, your guess is correct. Setting a relatively strict threshold is intended to improve data quality and reduce noise. During preliminary experiments, we explored other success-rate thresholds (e.g., 50% and 25%), but observed that these settings resulted in less stable training. We suspect this was due to the introduction of additional noise or potential reward hacking. Our SPC framework allows us to efficiently generate large amounts of raw data and obtain high-quality training data through filtering. Therefore, raising the threshold is relatively beneficial and harmless.
>
> **Q2. Solver selection**
>
> In fact, we have already considered both the performance and output style of the solvers in our selection process.
> 1. Our SPC can cover both weak and strong solvers  (see Table 3). Specifically, on AIME 2024, Llama-3.1-8B-Instruct achieves a maximum performance of 6.67%, while DeepSeek-R1-Distill-Qwen-7B reaches 73.3%, obviously demonstrating their performance gap.
> 2. Our SPC can generalize to different output styles (see Lines 283-285). We only used Llama-3.1-8B-Instruct, Qwen2.5-7B-Instruct, and Qwen2.5-32B-Instruct to collect the original solutions, which were then fed to the sneaky generator and critic for adversarial training. For testing, however, we applied SPC to DeepSeek-R1-Distill-Qwen-7B, a solver with strong reasoning capabilities and a completely different long CoT output style. On MATH500, compared to the PRM baseline result of 91.8%, using SPC to assist the R1-style solver achieved a performance of 94.0%, demonstrating an obvious improvement.

---

> ### Author Response · Authors · 2025-08-07
> **Follow-up on Our Rebuttal**
>
> Dear Reviewer pEmR,
>
> We would like to kindly follow up regarding our rebuttal, and also let you know that we have added some additional experiments and results in response to other reviewers’ concerns. We would appreciate it if you could review our responses and the new results as well. Your feedback would be very valuable to us. Thank you for your time and consideration!
>
> Best regards,
>
> The Authors

---

### Official Review · Reviewer_JDJJ · 2025-06-29

**Clarity:** 3
**Significance:** 4
**Originality:** 3
**Rating:** 5
**Confidence:** 3

**Summary:**

This paper proposes Self-Play Critic (SPC), an adversarial training framework designed to learn a more robust and effective process-based critic model. The core of SPC involves augmenting a standard critic model with a "Sneaky Generator." This generator is tasked with a specific adversarial goal: given a partially correct solution, it is trained to produce a plausible but incorrect subsequent step, aiming to deceive the critic. The critic, in turn, is trained to maximize its ability to discriminate between correct reasoning steps and the subtle errors introduced by the generator. Both the critic and the generator are optimized using reinforcement learning in a self-play loop. The authors validate the effectiveness of the resulting critic model on standard reward benchmarks and demonstrate improved performance with test-time scaling.

**Questions:**

Please refer to the weaknesses part.

**Ethical Concerns:**

["NO or VERY MINOR ethics concerns only"]

**Final Justification:**

The authors' rebuttal clearly addressed all my concerns especially for SPC's generalization abilitity to other models. Thus, I have raised my score from 4 to 5.

**Limitations:**

Please refer to the weaknesses part.

**Quality:**

3

**Strengths And Weaknesses:**

## Strengths

1. The proposed adversarial self-play approach is a novel contribution to the field of process-based feedback. By generating its own challenging training data, the method effectively improves the critic's ability to detect nuanced errors, moving beyond the limitations of static datasets.

2. The SPC framework helps to alleviate the well-known bottleneck of acquiring large-scale, high-quality human preference data for training process reward models. By synthetically generating difficult negative examples, it amplifies the utility of an initial seed dataset.

3. Clarity and Structure: The paper is well-written and logically structured, making the complex interplay between the generator and critic accessible and easy to follow.

## Weaknesses

1. A notable limitation is that the SPC framework still requires a pre-existing, human-annotated dataset (e.g., PRM800K) to bootstrap the training process. This dependency means that SPC does not entirely solve the foundational problem of data acquisition for PRMs but rather acts as a data amplifier and refiner. The quality and domain of the final critic are likely still heavily influenced by the initial seed data.

2. The experiments appear to be conducted with a specific choice of model architecture (Qwen) for the critic and generator. The paper would be more persuasive if it included ablation studies exploring the framework's performance with different underlying models (e.g., using Llama 3.1 for both components). This would help ascertain whether the benefits of SPC are generalizable across various model families and sizes or if they are dependent on a particular architectural setup.

3. SPC involves iterative adversarial training and reinforcement learning, which appears to be computationally intensive. The paper does not provide details on the training time, computational resources required, or the overall efficiency of the self-play loop. This information is crucial for assessing the practical viability and scalability of the SPC framework. It is recommended that the authors clarify these costs in the main text.

4. The results from the test-time scaling experiments raise potential concerns about the generalization capabilities of the learned critic. The performance improvement on the AIME dataset seems considerably smaller than on GSM8K. A plausible hypothesis is that this discrepancy arises because the bootstrap dataset (PRM800K) is collected from solutions to GSM8K problems.

---

> ### Author Rebuttal · Authors · 2025-07-31
>
> We are very pleased to hear that you consider SPC a novel contribution that alleviates the well-known bottleneck of data, and we appreciate your recognition of our efforts in making the complex interplay between the generator and critic accessible and easy to follow. We also address your concerns below.
>
> **W1. Human-annotated data**
>
> We did utilize a portion of human-annotated data (PRM800K) to perform a cold start for the model, which is a common training stage prior to RL training. It is reasonable to use these seed data, because the 7B model we use does not have sufficient initial capabilities as a sneaky generator or critic. If a more capable model were used, our framework could theoretically skip this cold start step and proceed directly to RL training. Directly enabling a stronger model to self-evolve is indeed a valuable direction to explore.
>
> Besides, in RL training phase, we only utilize our self-play framework to generate new data and do not rely on any human data. It is also worth noting that the PRM800K dataset consists of human annotations based on early GPT-4 outputs, and the resulting PRMs are thus tailored to the style of GPT-4 outputs. By continuously generating data through self-play, we successfully generalized our approach to multiple types of solvers, including DeepSeek-R1-Distill-Qwen-7B, Qwen2.5-32B-Instruct, and Llama-3.1-8B-Instruct. Our method makes full use of existing publicly available data and mitigates the influence of the initial data to some extent.
>
> **W2. Training different models**
>
> We chose the current base model, Qwen2.5-7B-instruct, because it has relatively strong capabilities in this model size. Some contemporaneous works [1, 2] have also made similar choices. In addition, we have also verified that our SPC can effectively improve the performance of different solver types (Llama3.1, Qwen2.5, and Deepseek-R1), which to some extent demonstrates the generalization ability of our framework. Thank you for your comments, and we will also try to verify the generalizability of our SPC by training other model families, such as Llama.
>
> [1] Tang et al. "RefCritic: Training Long Chain-of-Thought Critic Models with Refinement Feedback." arXiv 2507.
>
> [2] Yang et al. "DeepCritic: Deliberate Critique with Large Language Models." arXiv 2505.
>
> **W3. Details of Experimental Costs**
>
> During the data collection phase, we used 8 H20 GPUs to collect offline data, and each round of self-play took approximately 8 hours. For training, our final critic model used about 13.2K data samples for offline RL and required 3 epochs of training. When we used a cluster with 32 GPUs, training a single model takes about 2.5 hours, whereas using 8 GPUs takes about 8 hours.  Besides, our framework only needs to use offline RL, which requires less computational resources and time compared to recent work using online RL. Overall, the resources required by our SPC framework are moderate.
>
> **W4. The generalization capabilities of the critic model**
>
> Thank you for your thoughtful consideration. First, we would like to kindly point out a small typo in your comments: our experiments were conducted on MATH500, not GSM8K. We have also noticed the difference in performance gains of SPC on MATH500 and AIME2024. However, we believe that this does not necessarily indicate weaker generalization on AIME. In fact, the experimental results on Qwen2.5-32B-Instruct show the opposite. The result of Self-Consistency + SPC on MATH500 is 85.2%, which outperforms the 80.8% achieved by Self-Consistency + Math-Shepherd. On AIME2024, their performances are 23.3% and 13.3%, respectively. Obviously, when assisting Qwen2.5-32B-Instruct in problem solving, SPC brings a more significant performance improvement on AIME2024.
>
> We also agree with your hypothesis. In our experiments, the training dataset we are using is not very large in scale or high in difficulty. Recent work [3,4] has shown that increasing the difficulty and scale of the data is beneficial for further improving model performance. We believe that training our framework on larger-scale and more challenging data will lead to better results on more challenging test sets such as AIME.
>
> [3] He et al. "DeepMath-103K: A Large-Scale, Challenging, Decontaminated, and Verifiable Mathematical Dataset for Advancing Reasoning." arXiv 2025.
>
> [4] Hu, Jingcheng, et al. "Open-reasoner-zero: An open source approach to scaling up reinforcement learning on the base model." arXiv 2025.

---

> ### Author Response · Authors · 2025-08-03
>
> Thank you for your valuable suggestions. In response to your request, we spent some time adapting our training code and environment to be compatible with Llama-3.1, and have obtained some preliminary results, as shown in the table below. Specifically, we used the same data to first cold-start a critic based on **Llama-3.1-8B-Instruct**, and then collected data through adversarial self-play with a sneaky generator for subsequent RL training. Due to time constraints, we conducted only one round of adversarial training. The experimental results show that our Llama-based critic also achieves significant performance improvements across three benchmarks.
>
> | Methods       | ProcessBench | PRM800K | DeltaBench |
> |---------------|:--------------:|:---------:|:------------:|
> | Prompting     | 56.2         | 51.9    | 49.1       |
> | SPC (Round 0) | 61.1         | 58.9    | 55.8       |
> | SPC (Round 1) | 66.2         | 62.0    | 58.4       |
>
> Table: "Prompting" refers to the performance of the original Llama-3.1-8B-Instruct, while "round 0" and "round 1" represent the performance of our SPC (based on Llama-3.1-8B-Instruct) after cold start and after one round of self-play, respectively.
>
> These results validate that SPC can well adapt to other model family, verifying the generality of our proposed algorithm. We hope that these results sufficiently address your concerns.

---

> > ### Comment · Reviewer_JDJJ · 2025-08-04
> > **Response to rebuttal**
> >
> > Thanks to the author for Llama-3.1 experiments! It addressed my concerns and I have revised my score accordingly.

---

> > > ### Author Response · Authors · 2025-08-04
> > >
> > > We are very glad to hear that your concerns have been addressed and that your rating score has been revised. Thank you again for your review work!

---

### Official Review · Reviewer_czJQ · 2025-06-30

**Clarity:** 3
**Significance:** 4
**Originality:** 4
**Rating:** 5
**Confidence:** 4

**Summary:**

This paper proposes Self-Play Critic (SPC), a novel framework for training a step-level verifier for CoT reasoning. SPC uses an adversarial self-play game between two roles: a sneaky generator that injects subtle reasoning errors, and a critic that identifies them. Through reinforcement learning on the outcomes of these interactions, both models co-evolve. Experiments across reasoning benchmarks (ProcessBench, PRM800K, DeltaBench) demonstrate the superior performance of SPC.

**Questions:**

The evaluation process on PRM800K and DeltaBench seems to deviate from their original settings (as noted in lines 244–246).  Could the authors clarify the specific modifications made to the evaluation process and justify how these changes may impact the comparability of previous PRM methods?

**Ethical Concerns:**

["NO or VERY MINOR ethics concerns only"]

**Final Justification:**

No more rebuttal is made by authors. I maintain my original assessment.

**Limitations:**

yes

**Quality:**

4

**Strengths And Weaknesses:**

Strengths:
- The self-play setup at the step-level is original and well-motivated, filling a clear gap in current PRMs.

- The paper includes solid and comprehensive experiments across several benchmarks. SPC shows consistent improvements across all.

- The paper is well written.


Weakness:
In my view, the paper does not have any major fundamental issues. The only issue is that ORM baselines should also be considered in BoN-style evaluation, such as [1]

[1] Generative Verifiers: Reward Modeling as Next-Token Prediction

---

> ### Author Rebuttal · Authors · 2025-07-31
>
> Thank you for considering our approach to be original and well-motivated, with solid and comprehensive experiments, and for your positive comments on the writing of the paper. We have addressed your questions below.
>
> **W1. ORM baseline**
>
> Thank you for your suggestion. We have checked several representative generative ORMs [1, 2, 3], including the Generative Verifiers work you mentioned. Unfortunately, none of them have released their trained model weights. Therefore, we found another open-source ORM on Hugging Face, which was trained using Deepseek's data. We compare the performance of this new ORM baseline with other baselines as well as our proposed SPC. For example, on Qwen2.5-32B-Instruct solver, the test results are as follows:
>
> | Methods    | MATH500 | AIME2024  |
> |------------|:---------:|:-------------------:|
> | ORM     | 78.2   |   13.3    |
> | Math-Shepherd     | 78.8       | 13.3  |
> |Qwen2.5-Math-7B-PRM800K | 82.8 | 16.7 |
> | Self-Consistency + ORM | 82.4 | 16.7 |
> | Self-Consistency + Math-Shepherd | 80.8 |13.3 |
> |Self-Consistency + Qwen2.5-Math-7B-PRM800K| 84.6 | 16.7 |
> | SPC     | 83.0   | 16.7 |
> |Self-Consistency + SPC | 85.2 | 23.3 |
>
> [1] Zhang et al. "Generative Verifiers: Reward Modeling as Next-Token Prediction." ICLR 2025.
>
> [2] Ye et al. "Beyond Scalar Reward Model: Learning Generative Judge from Preference Data." arXiv 2024.
>
> [3] Mahan et al. "Generative Reward Models." arXiv 2024.
>
> **Q1. Evaluation settings**
>
> To better reflect the practical effectiveness of PRMs/Critic models in test-time search, we adopt a different evaluation setting. This is because, during test-time search, it is more important for the critic model to analyze the current step and promptly correct potential errors in this step. Thus, only the current step and the preceding reasoning steps (partial solution) will be used as input, rather than analyzing each step in the entire solution from the beginning. In this setting, PRMs are inevitably affected due to their inherent limitations of typically providing step-level scores on the complete solutions. Therefore, on both PRM800K and DeltaBench, we further compare against a wide range of baselines that prompt powerful LLMs as critic models. The results demonstrate that our SPC significantly outperforms these baselines.

---

> > ### Comment · Reviewer_czJQ · 2025-08-01
> > **Response by Reviewer**
> >
> > Thank you for your response. I will maintain my original score and assessment.

---

> > > ### Author Response · Authors · 2025-08-01
> > >
> > > Thank you for your positive comments on our work and your prompt response!

---

### Official Review · Reviewer_d2Hw · 2025-07-06

**Clarity:** 3
**Significance:** 3
**Originality:** 3
**Rating:** 4
**Confidence:** 1

**Summary:**

The key idea of this paper is intuitive: leveraging a self-play critic to improve the quality of the LLM reasoning process. The experiments show substantial improvement on several benchmarks, even after just a couple of iterations. The paper is well written and easy to follow.

**Questions:**

The major question is that do you have more insights on the rounds of self-play. What's would happen if it continously increase?

**Ethical Concerns:**

["NO or VERY MINOR ethics concerns only"]

**Limitations:**

yes

**Quality:**

3

**Strengths And Weaknesses:**

The idea of this paper is intuitive and it should be a direction to explore  to further improve the LLM reasoning. Though the submission is not in your area, I expect that there should be some related works that not well discussed in the paper.

---

> ### Author Rebuttal · Authors · 2025-07-31
>
> We thank you for your comments that our idea is intuitive, our experiments show substantial improvement, and the paper is well written and easy to follow. We have addressed your question below.
>
> **More insights on the rounds of self-play**
>
> Thank you for your question. We believe that the essence of adversarial training in our SPC lies in fully exploring and mining the potential erroneous reasoning steps that may arise when attempting to solve certain problems, and using this data to train the sneaky generator and critic model. Considering the balance between data collection efficiency and performance gains, conducting 2-3 rounds of iteration is appropriate to fully utilize the problems in the MATH dataset. In addition, we think that the rounds of iterations may also be related to the difficulty of the problems. For more challenging mathematical problems or other complex tasks (e.g., training general agents), it may be worthwhile to conduct more rounds of exploration, as we could continuously generate more diverse data from difficult problems. We will provide more insights and analysis in our revised version.

---

### Decision · Program_Chairs · 2025-09-17

**Decision:**

Accept (poster)

**Comment:**

The paper introduces Self-Play Critic (SPC), a novel framework that trains a critic model to evaluate step-by-step LLM reasoning. The core contribution is an adversarial self-play game where a "sneaky generator" model learns to produce subtle reasoning errors, and a "critic" model evolves its ability to detect them. This process uses reinforcement learning from game outcomes, which reduces the need for extensive, step-level human annotations.

On the positive side, the reviewers found that the paper presents an original and well-motivated approach to a significant problem. Reviewers (czJQ, JDJJ, oHJP) described the idea as novel, intuitive, and a useful contribution. They noted that the framework effectively alleviates the data acquisition bottleneck for process reward models by generating its own challenging training data. The reviewers also found the experimental results to be solid and comprehensive, showing that SPC consistently improves performance across multiple benchmarks and outperforms baselines.

The reviewers identified areas of improvement, primarily concerning the scope and generalizability of the proposed method. One reviewer (pEmR) noted that the experiments were confined to the domain of mathematical reasoning, which raises questions about whether the "sneaky generator" strategy would be effective for more diverse tasks like coding or assistant tasks. Another reviewer (JDJJ) initially questioned if the benefits were specific to the Qwen model architecture, though this concern was later addressed by the authors with new experiments. Other points included the dependency on an initial human-annotated dataset to start the process and an initial lack of detail on the computational cost of the training loop. I think this is an important point to make more clear in the paper, since the initial annotated dataset deviates from the "Self-Play" claims in the abstract.

Based on these reviews, I recommend accepting this paper. The reviewers were excited about the paper's novel and intuitive framework for improving reasoning critics without extensive human annotation. They highlighted the originality of the step-level self-play setup, its ability to address the data acquisition bottleneck, and its strong empirical performance on several challenging benchmarks. The authors were responsive during the discussion period and provided additional experiments on a different model family (Llama-3.1), which resolved the concerns led to raised their scores. I expect the authors to incorporate the detailed feedback from the reviewers into the final version of the paper, especially the new results, and some clarification on the use of annotated data in addition to self-play.